# THE GEOMETRICAL AND TOPOLOGICAL SIGNATURES OF TRANSFORMERS

**Asif Khan**
Harvard Medical School, USA
asif.khan@hms.harvard.edu

## ABSTRACT

We propose a topological framework to analyze the layerwise evolution of transformer representations by modeling attention heads as Markov kernels on a token metric space. This formulation admits a Wasserstein-1 ($W_1$) lifting where coarse Ollivier-Ricci curvature provides quantitative bounds on the action of the induced operator. A positive curvature implies layerwise Wasserstein contraction while negative implies expansion. To connect these statements to practice, we introduce a reproducible probe that estimates robust curvature lower quantiles, directly tests contraction on random measures in $W_1$, and tracks layerwise topological simplification using persistent homology on diffusion-induced distances. In pretrained GPT-2 and GPT-2-medium models, we observe a depthwise transition toward more contractive support, with shrinking ($H_1$) lifetimes and persistence of a coarse ($H_0$) skeleton.

## 1 INTRODUCTION

Transformer layers combine token mixing through self-attention with features updated through residual blocks (Vaswani et al., 2017). Despite this simplicity, it is unclear what geometric structure these operations impose across depth (Ethayarajh, 2019; Dong et al., 2021). Which notion of distance do transformers contract, preserve, or distort? In this work, we analyze the geometry through a metric and probabilistic lens focusing on what the architecture enforces (e.g., causal masking) and what is learned by the parameters.

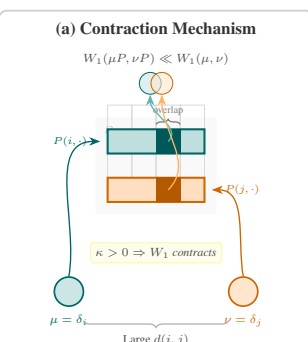 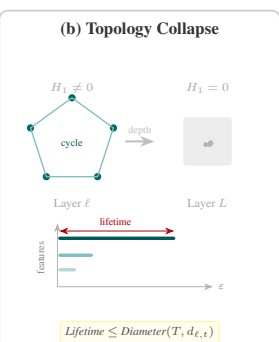 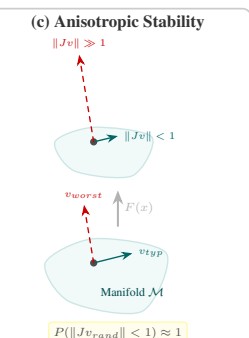

Figure 1: **Geometric Contraction in Transformers.** (a) **Curvature** Attention maps distinct tokens $i, j$ to a common target column, strictly contracting the $W_1$ distance between their measures. (b) **Topology.** Contraction reduces diffusion-space diameter and shortens the persistence of loop-like topological features. (c) **Stability.** Worst-case directions (red, dashed) may expand, they are orthogonal to the data manifold, typical directions (green) remain contractive.

We make this precise using the coarse Ollivier–Ricci curvature (Ollivier, 2009). Intuitively, this curvature measures whether random walks started at nearby points tend to move towards each other. So when two tokens attend to similar contexts, their distributions overlap, resulting in positive curvature. A lower bound on positive curvature implies strong overlap between local neighborhoods,

which guarantees that the operator contracts the $W_1$ distance relative to the chosen ground metric. In our setting, we interpret this contraction as a form of geometric simplification meaning fin-scale distinctions become less persistent across depth while coarser structure remain. We validate this hypothesis directly by computing contraction ratios using robust low quantiles which also isolates consistent signal within heavy-tailed curvature statistics.

However, contraction has a risk of collapsing everything to a single point. To prevent this problem residual connections act as a structural anchor as long as the residual branch is Lipschitz with constant $L_R < 1$ on the data manifold, the overall block remains bi-Lipschitz, preventing topological collapse and preserving information (Thm. 2). We empirically observe that spectral norms of residual Jacobians frequently exceed one, implying local expansivity. But, in high-dimensional spaces, this worst-case bound is an unreliable proxy for stability due to the concentration of measure phenomenon. The expansive directions corresponding to the largest singular values which occupy negligible volume on the hypersphere. Therefore, what matters is a typical behavior, and typical directions contract (Thm. 3). We link this contraction to topology by showing the intrinsic diameter of the diffusion geometry shrinks which bounds the lifetimes of persistent homology cycles (Thm. 4).

Overall, this paper presents an evaluation framework for analyzing representation geometry of LLMs through, (i) a multi-metric curvature estimator with robust lower-quantile bounds; (ii) direct contraction tests on measures that validate curvature-based predictions; (iii) attention ablations that isolate learned geometric structure; (iv) probabilistic on-manifold stability diagnostics that explain worst-case Lipschitz violations; and (v) topological and causal-graph checks that connect diffusion geometry to architectural constraints.

## 2 GEOMETRIC CONTRACTION VIA COARSE RICCI CURVATURE

**Notations.** $(T = 1, \ldots, N)$ is the token index set, $d(.,.)$ is a ground metric on tokens, $P^{(\ell,h)}$ is the row-stochastic attention kernel for head $h$ at layer $\ell$. Then, $(W_1)$ is the Wasserstein-1 distance induced by $d(.,.)$, $(\kappa(i,j) = 1 - \frac{W_1(P(i,\cdot),P(j,\cdot))}{d(i,j)})$ is the coarse Ollivier–Ricci curvature, and $d_{\ell,t}(i,j) = W_1(\delta_i P^{(\ell)t}, \delta_j P^{(\ell)t})$ is the diffusion distance used for persistent homology (PH). We refer readers to Appendix S2 for detailed notations, preliminaries, and proofs.

### 2.1 CURVATURE LOWER BOUNDS IMPLY $W_1$ CONTRACTION

The following theorem is standard in the optimal-transport (Ollivier, 2009; Joulin and Ollivier, 2010) that becomes relevant here because $P^{(\ell)}$ is directly observable from pretrained models and the inequality is directly testable.

**Theorem 1** (Coarse curvature implies $W_1$ contraction). *Let $(\mathcal{T}, d)$ be a finite metric space and let $P$ be a Markov kernel on $\mathcal{T}$ with $\kappa_* = \inf_{i \neq j} \kappa(i,j)$ as defined in Section S2. Then for all $\mu, \nu \in \Delta(\mathcal{T})$, $W_1(\mu P, \nu P) \leq (1 - \kappa_*) W_1(\mu, \nu)$. Consequently, for all integers $t \geq 1$,*

$$W_1(\mu P^t, \nu P^t) \leq (1 - \kappa_*)^t W_1(\mu, \nu). \tag{1}$$

**Corollary 1** (Layerwise products). *If $P_1, \ldots, P_L$ are Markov kernels with curvature lower bounds $\kappa_{*,1}, \ldots, \kappa_{*,L}$, then $W_1(\mu P_1 \cdots P_L, \nu P_1 \cdots P_L) \leq \left( \prod_{\ell=1}^{L} (1 - \kappa_{*,\ell}) \right) W_1(\mu, \nu)$.*

Thm. 1 is a worst-case guarantee using $\inf \kappa$, if curvature values are drawn from a distribution $\mathcal{D}$ and $\kappa_{lb}$ is the $q$-quantile, then for a uniformly random pair $(i,j)$, the contraction ratio satisfies $W_1(P(i,\cdot), P(j,\cdot))/d(i,j) \leq 1 - \kappa_{lb}$ with probability at least $1 - q$. This reframes the bound from deterministic worst-case to high-probability typical-case. The direct contraction tests in Section S8 validate that high-quantile contraction ratios track $1 - \kappa_{lb}$ even when the sampled infimum is noisy.

### 2.2 RESIDUAL CONNECTIONS PREVENT TOTAL FEATURE-SPACE COLLAPSE.

Next, we show a sufficient condition to ensure that a residual map cannot collapse all on-data distances.

**Assumption 1** (On-data Lipschitz residual). *There exists $L_R \geq 0$ such that for all $u, v \in \mathcal{X}$, $\|R(u) - R(v)\| \leq L_R \|u - v\|$.*

**Theorem 2** (Residual bi-Lipschitz bound)**.** *Let $F(u) = u + R(u)$ on $\mathcal{X}$ and assume Assumption 1. Then for all $u, v \in \mathcal{X}$, $(1 - L_R) \|u - v\| \leq \|F(u) - F(v)\| \leq (1 + L_R) \|u - v\|$. In particular, if $L_R < 1$ then $F$ is injective on $\mathcal{X}$ (no total collapse on-data).*

We estimate on-data residual Lipschitz behavior empirically both from sampled pairwise ratios $\Lambda_\ell(u, v) = \|R_\ell(u) - R_\ell(v)\| / \|u - v\|$ and from Jacobian proxies ( $\|J_{R_\ell}\|_2$ and random-direction JVP statistics). However, spectral norm captures only the most expansive direction. In high dimension, this direction occupies a negligible fraction of the sphere. The relevant question for on-distribution behavior is, "what happens along typical directions?"

**Theorem 3** (Typical-direction stability from Frobenius control)**.** *Let $J \in \mathbb{R}^{d \times d}$ and $s \sim \mathrm{Unif}(\mathbb{S}^{d-1})$. Then $\mathbb{E}[\|Js\|^2] = \frac{\|J\|_F^2}{d}$, and for any $t > 0$,*

$$\mathbb{P}(\|Js\| \geq t) \leq \frac{\|J\|_F^2}{d\, t^2}.$$

*In particular, $\mathbb{P}(\|Js\| \geq 1) \leq \|J\|_F^2 / d$.*

**Corollary 2** (Probabilistic residual contraction)**.** *Let $p_\ell$ denote the distribution of hidden states at layer $\ell$, and suppose the normalized pairwise differences $(u - v)/\|u - v\|$ for $u, v \sim p_\ell$ are approximately uniform on $\mathbb{S}^{d-1}$. Define $\rho_\ell^2 = \mathbb{E}_{h \sim p_\ell}[\|J_{R_\ell}(h)\|_F^2]/d$. Then the on-distribution directional Lipschitz satisfies $\mathbb{P}_{u,v}(\Lambda_\ell(u, v) < 1) \geq 1 - \rho_\ell^2$, where $\Lambda_\ell(u, v) = \|R_\ell(u) - R_\ell(v)\| / \|u - v\|$.*

## 2.3 FROM CONTRACTION TO PERSISTENCE BOUNDS

Intuitively, repeated contraction causes distinct token neighborhoods to become less distinguishable in diffusion distance. PH provides a multiscale summary of what geometric structure survives that contraction. For example, long $H_1$ lifetime indicates robust loop-like structure in the induced token geometry, whereas shrinking lifetime indicate progressive simplification. Next, we connect metric contraction to topological signatures and establish an upper bound on persistence lifetimes via diameter control.

**Theorem 4** (Curvature controls persistence lifetimes)**.** *Let $(\mathcal{T}, d_{\ell,t})$ be the token set equipped with diffusion distance induced by $P^{(\ell)}$ at layer $\ell$ (Definition 1). Assume $\kappa_{\ell,*} > 0$ is a global lower bound on the Ollivier–Ricci curvature of $P^{(\ell)}$ with respect to the ground metric $d$. Then,*

*(i) Diameter contraction. For all $t \geq 1$, $\mathrm{diam}(\mathcal{T}, d_{\ell,t}) \leq (1 - \kappa_{\ell,*})^{t-1} \mathrm{diam}(\mathcal{T}, d_{\ell,1})$. Equivalently, for all $t \geq 1$, $\mathrm{diam}(\mathcal{T}, d_{\ell,t}) \leq (1 - \kappa_{\ell,*})^t \mathrm{diam}(\mathcal{T}, d)$.*

*(ii) Lifetime bound. In the Vietoris–Rips (VR) filtration of $(\mathcal{T}, d_{\ell,t})$, every $H_k$ feature with $k \geq 1$ has lifetime at most $\mathrm{diam}(\mathcal{T}, d_{\ell,t})$.*

**Corollary 3** (Exponential suppression of $H_1$ lifetimes)**.** *Under $\kappa_{\ell,*} > 0$, the maximum $H_1$ lifetime in the Rips persistence diagram of $(\mathcal{T}, d_{\ell,t})$ satisfies*

$$\max \mathrm{lifetime}(H_1) \leq (1 - \kappa_{\ell,*})^{t-1} \cdot \mathrm{diam}(\mathcal{T}, d_{\ell,1}) \leq (1 - \kappa_{\ell,*})^t \cdot \mathrm{diam}(\mathcal{T}, d).$$

According to stability theorem for persistence (Cohen-Steiner et al., 2007) if two metrics $d, d'$ on the same finite set satisfy $\|d - d'\|_\infty \leq \varepsilon$, then the bottleneck distance between their persistence diagrams satisfies $d_B(\mathrm{Dgm}(d), \mathrm{Dgm}(d')) \leq \varepsilon$. Combined with contraction bounds, this implies diagrams cannot change by more than the corresponding metric perturbation in $\|\cdot\|_\infty$.

## 3 EXPERIMENTAL RESULTS

We now carry out a set of empirical validation to test whether the curvature contraction relationship and the associated topological trends hold in practice. We analyze representations of GPT-2 ($L = 12$ layers) and GPT-2-medium ($L = 24$ layers), using 50 natural-text prompts (sequence length $L = 128$), estimate Ollivier–Ricci curvature via MC pair sampling (reporting the robust lower bound $\hat{\kappa}_{\ell,\mathrm{lb}}$ as the 5th percentile), and test direct $W_1$ contraction by sampling random measures $\mu, \nu \sim \mathrm{Dirichlet}(\alpha)$ and measuring the ratio $r_\ell = \frac{W_1(\mu P_\ell, \nu P_\ell)}{W_1(\mu, \nu)}$ (Algorithm 1). We report the median contraction ratio across trials, together with layerwise trends. Details on metrics and experiment setup

---

**Algorithm 1** Curvature–Contraction–Topology Probe (single prompt)

---

**Require:** Model $\mathcal{M}$, tokenized prompt of length $N$, ground metric $d$, diffusion time $t$
**Require:** Pair sample size $M$ (default 5,000), Dirichlet trials $R$ (default 200)
**Ensure:** Per-layer summaries: $\widehat{\kappa}_{\text{lb}}^{(\ell)}$, contraction quantiles, PH summaries
  1: Run $\mathcal{M}$ with attention saved. For each layer $\ell$, form row-stochastic $P^{(\ell)}$ (either head-averaged or per-head $P^{(\ell,h)}$).
  2: Sample $M$ pairs $S = \{(i_m, j_m)\}$ uniformly from $\{(i,j) : i < j\}$.
  3: **for** each layer $\ell$ **do**
  4:    **Curvature:** compute $\widehat{\kappa}^{(\ell)}(i,j)$ for $(i,j) \in S$ and set $\widehat{\kappa}_{\text{lb}}^{(\ell)} = \text{Quantile}_{0.05}(\{\widehat{\kappa}^{(\ell)}(i,j)\})$.
  5:    **Direct contraction:** for $r = 1, \ldots, R$, sample $(\mu_r, \nu_r) \sim \text{Dirichlet}(\beta\mathbf{1})$, compute $\alpha_r^{(\ell)} = \frac{W_1(\mu_r P^{(\ell)}, \nu_r P^{(\ell)})}{W_1(\mu_r, \nu_r) + \varepsilon}$, and record $\text{median}(\alpha_r^{(\ell)})$, $\text{Quantile}_{0.9}(\alpha_r^{(\ell)})$.
  6:    **Topology:** compute diffusion distance matrix $D_{ij}^{(\ell)} = W_1(\delta_i P^{(\ell)t}, \delta_j P^{(\ell)t})$, run VR persistent homology on $(\mathcal{T}, D^{(\ell)})$ and record max lifetimes in $H_0, H_1$.
  7: **end for**

---

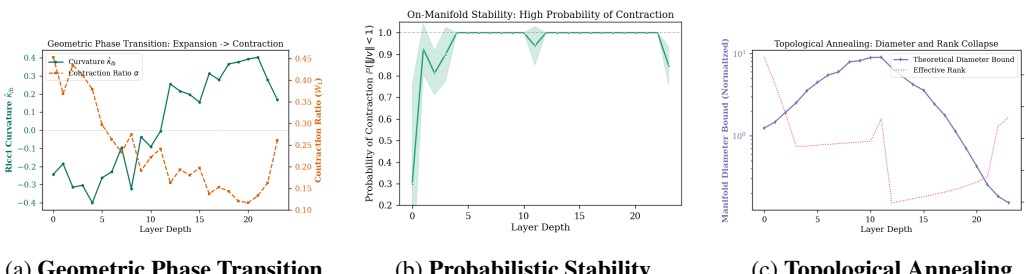

| (a) **Geometric Phase Transition** | (b) **Probabilistic Stability** | (c) **Topological Annealing** |

Figure 2: (**a**) In the tested GPT-2 models, curvature becomes more positive with depth while the empirical $(W_1)$ contraction ratio decreases, consistent with the theoretical curvature–contraction connection. (**b**) Residual updates can be spectrally expansive in worst-case directions, yet remain contractive with high probability on the observed data manifold. (**c**) As contraction accumulates across depth, the diffusion-space diameter shrinks and the persistence of loop-like topological structure decreases.

are included in Appendix S6. The results support the geometric and topological claims and show a distinct two-phase depthwise pattern. An early expansion phase associated with negative curvature, followed by a later contraction phase in which the induced topology becomes simpler (Figure 2).

The layerwise correlation between curvature and contraction is $\rho = -0.83$ for GPT-2 ($p < 10^{-3}$) and $\rho = -0.77$ for GPT-2-medium ($p < 10^{-5}$), which confirms that measured contraction tracks curvature geometry. Next, we look at the topological consequences of this curvature using the diameter bound $D_L \leq$

Table 1: Summary statistics (positional metric, trained attention).

| Model | $\bar{\kappa}_{lb}$ | $\bar{r}$ | $\mathbb{P}(\Lambda < 1)$ | Forbidden |
|---|---|---|---|---|
| GPT-2 | 0.030 | 0.291 | 0.861 | 0.000 |
| GPT-2-medium | 0.496 | 0.150 | 0.931 | 0.000 |
| BERT | −0.994 | 0.385 | 0.994 | 0.570 |

$D_0 \prod_{l=0}^{L}(1 - \kappa_l)$. The diameter initially expands during feature extraction, before collapsing exponentially in the contraction phase to $< 20\%$ of its original scale (Figure 2c). This annealing leads to a suppression of topological cycles which implies complex homological features created in early layers are geometrically crushed in deeper layers.

As expected from causal masking, the decoder-only attention graphs contains no directed cycles across layers. Consistent with the structural prediction, the measured forbidden attention mass is effectively zero. This empirically shows that autoregressive masking annihilates cyclic dependencies.

**Ablation.** To isolate which properties of attention produce the observed geometry, we replace each learned attention matrix with controlled alternatives (preserving the causal mask): *Uniform* ($P_{ij} \propto 1$ on valid positions), *Positional decay* ($P_{ij} \propto e^{-|i-j|/\tau}$), and *Permuted* (shuffle key indices within each causal row, preserving row entropy but destroying specific $Q$–$K$ alignment).

Uniform attention results in maximal curvature and complete contraction representing trivial smoothing rather than learned structure. Positional decay shows moderate curvature but weak contraction. Neither baseline reproduces the trained model's depth-dependent evolution (Figure 2).

The positional decay baseline is especially informative. Although, it imposes a strong locality bias, it produces weaker contraction ($\bar{r} = 0.564$) than the trained attention model ($\bar{r} = 0.291$). This suggests that the geometric signature depends on learned, content-dependent transport in the tested model.

Table 2: Attention ablations (GPT-2-medium, positional metric).

| Condition | $\bar{\kappa}_{\mathrm{lb}}$ | $\bar{r}$ | $P(\Lambda < 1)$ |
|---|---|---|---|
| Trained | **0.496** | **0.150** | **0.931** |
| Uniform | 0.285 | 0.600 | 0.466 |
| Pos-decay | 0.271 | 0.605 | 0.407 |
| Permuted | −0.853 | 0.506 | - |

**On-manifold stability of residual blocks** Finally, we test the stability claim at the level of residual updates, distinguishing worst-case (global) behavior from *on-manifold* behavior over actual token representations. For each layer, we estimate $L_R(h_i, h_j) = \frac{\|R_\ell(h_i) - R_\ell(h_j)\|}{\|h_i - h_j\|}$, over randomly sampled token pairs $(i, j)$ drawn from the empirical layer distribution.

**Residual updates are typically contractive on-manifold in mid-layers.** While worst-case Lipschitz estimates can exceed 1, the on-manifold distribution concentrates below 1 throughout most of the network. For GPT-2, the average $\mathbb{P}(\Lambda < 1)$ across layers is $86.1\%$; for GPT-2-medium, this increases to $93.1\%$ (Table 1). Mid-layers are near-certainly contractive, with the main exceptions occurring at embedding and final layers (Figure 2b).

**Typical vs worst-case directions.** The gap between median $\Lambda$ and 95th-percentile $\Lambda$ confirms anisotropic stability (Figure 2b) worst-case spectral directions are not representative of on-manifold behavior. This validates Theorem 3's prediction that Lipschitz instability is a concentration phenomenon confined to rare directions.

## 4 CONCLUSION

We introduced a geometric and topological probe for studying layerwise transport in transformer attention by viewing attention heads as Markov kernels on a token metric space. The framework links coarse Ollivier–Ricci curvature to Wasserstein contraction and connects that contraction to shrinking persistent topological structure in the induced diffusion geometry. In the pre-trained GPT-2 models, we observe a transition from low-curvature transport (preserving geometric detail) to high-curvature contraction (compressing the measure space). Furthermore, our analyses show expansive instability is confined to rare, off-manifold directions. This implies that network architectural design should target the stability of the typical set, regulating the rate of topological simplification to balance semantic convergence with feature retention.

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

## SUPPLEMENTARY MATERIAL

## S1 NOTATIONS

We adopt standard notation on optimal transport, Markov chains (Villani, 2009; Ollivier, 2009; Paulin, 2016), and persistent homology (PH)(Edelsbrunner and Harer, 2010; Chazal et al., 2016). We explicitly define any specialized notation where our usage differs from above references.

| Symbol | Meaning |
|---|---|
| $N$ | Sequence length after padding/truncation. |
| $\mathcal{T} = \{1, \ldots, N\}$ | Token index set (positions). |
| $i, j \in \mathcal{T}$ | Token indices / positions. |
| $L$ | Number of layers. |
| $\ell \in \{0, \ldots, L-1\}$ | Layer index. |
| $H$ | Number of attention heads. |
| $h \in \{1, \ldots, H\}$ | Head index. |
| $d(\cdot, \cdot)$ | Ground metric on $\mathcal{T}$ (e.g., positional, embedding, cosine). |
| $d_{\mathrm{pos}}(i, j)$ | Normalized positional metric $|i - j|/(N-1)$. |
| $\Delta(\mathcal{T})$ | Probability simplex over tokens: $\{\mu \in \mathbb{R}_+^N : \sum_i \mu_i = 1\}$. |
| $\mu, \nu \in \Delta(\mathcal{T})$ | Distributions over token indices. |
| $\Pi(\mu, \nu)$ | Set of couplings (transport plans) between $\mu$ and $\nu$. |
| $W_1^d(\mu, \nu)$ | 1-Wasserstein distance induced by ground metric $d$. |
| $\delta_i$ | Dirac measure at token $i$ (one-hot distribution). |
| $P^{(\ell, h)}$ | Row-stochastic attention kernel (Markov kernel) at layer $\ell$, head $h$. |
| $P^{(\ell)}$ | Head-averaged attention kernel at layer $\ell$ (if averaging). |
| $P(i, \cdot)$ | Row distribution of kernel $P$ at token $i$. |
| $\mu P$ | Pushforward of $\mu$ by kernel $P$: $(\mu P)_j = \sum_i \mu_i P_{ij}$. |
| $\kappa(i, j)$ | Ollivier–Ricci curvature along pair $(i, j)$, $1 - \frac{W_1(P(i,\cdot), P(j,\cdot))}{d(i,j)}$. |
| $\kappa_*$ | Global curvature lower bound $\inf_{i \neq j} \kappa(i, j)$. |
| $\widehat{\kappa}_{\ell, \mathrm{lb}}$ | Robust curvature lower-quantile estimator at layer $\ell$ (e.g., $q = 0.05$). |
| $t$ | Diffusion time (number of kernel steps). |
| $d_{\ell, t}(i, j)$ | Diffusion distance $W_1(\delta_i P^{(\ell)t}, \delta_j P^{(\ell)t})$. |
| $\mathrm{diam}(\mathcal{T}, d)$ | Diameter $\max_{i,j} d(i, j)$. |
| $\mathrm{VR}_\varepsilon(\mathcal{T}, d)$ | Vietoris–Rips complex at scale $\varepsilon$ from metric space $(\mathcal{T}, d)$. |
| $H_k$ | $k$-th homology group (persistent homology when tracked across $\varepsilon$). |
| $\max(\mathrm{death} - \mathrm{birth})$. | |
| $G = (V, E)$ | Directed attention graph (after masking/thresholding). |
| $\omega : E \to \mathbb{R}$ | Edge-flow on a directed graph. |
| $u, v \in \mathcal{X}$ | Hidden states restricted to the data manifold $\mathcal{X} \subset \mathbb{R}^d$. |
| $F(u) = u + R(u)$ | Residual map with residual branch $R$. |
| $L_R$ | (On-data) Lipschitz constant for $R$: $\|R(u) - R(v)\| \leq L_R \|u - v\|$. |
| $\Lambda_\ell(u, v)$ | On-data directional Lipschitz statistic $\|R_\ell(u) - R_\ell(v)\|/\|u - v\|$. |
| $J_{R_\ell}(u)$ | Jacobian of residual branch at layer $\ell$ evaluated at $u$. |
| $\|J\|_2, \|J\|_F$ | Spectral norm and Frobenius norm of a Jacobian $J$. |
| $r_{\mathrm{eff}}(J)$ | Effective rank $\|J\|_F^2 / \|J\|_2^2 \in [1, d]$. |
| $\rho_\ell^2$ | Frobenius-per-dimension scale: $\mathbb{E}\|J_{R_\ell}\|_F^2 / d$. |

## S2 PRELIMINARIES

At layer $l$, the representation is a function $X^{(l)} : \mathcal{T} \to \mathbb{R}^d$, which we view as a vector-valued 0-form on the token graph.

**Attention as a Markov kernel.** The self-attention mechanism induces a row-stochastic matrix $P^{(l)} \in [0, 1]^{N \times N}$, defined by $P_{ij}^{(l)} = \mathrm{softmax}(Q_i K_j^\top / \sqrt{d_k})$. We interpret $P^{(l)}$ as a Markov tran-

sition kernel on $\mathcal{T}$. In some experiments we use a *lazy* variant $\widetilde{P}^{(l)} := (1 - \alpha)I + \alpha P^{(l)}$ (with $\alpha \in (0, 1]$), which preserves stationarity while improving aperiodicity.

**Ground metric and Wasserstein lifting.** Fix a ground metric $d$ on $\mathcal{T}$ (e.g. normalized positional distance $d(i, j) = |i - j|/(N - 1)$, or other valid metrics used in Section S8). Let $W_1$ denote the 1-Wasserstein distance on $\Delta(\mathcal{T})$ induced by $d$.

**Coarse curvature.** The Ollivier coarse Ricci curvature of $P$ along $(i, j)$ is

$$\kappa(i, j) \;=\; 1 - \frac{W_1\big(P(i, \cdot), P(j, \cdot)\big)}{d(i, j)}, \qquad i \neq j, \tag{2}$$

and we write $\kappa_* = \inf_{i \neq j} \kappa(i, j)$ for the global lower bound.

**Directed masking and edge-flows.** The causal mask induces a directed graph $G = (V, E)$ with $V = \mathcal{T}$ and edges $(i \to j) \in E$ only when attention may flow. A decoder-only mask enforces $j \leq i$, so $G$ is a DAG. We will use the standard "circulation $\Leftrightarrow$ directed cycle" fact for such graphs.

**Residual blocks and on-data stability.** We model a residual sublayer as $F(u) = u + R(u)$ on the data manifold $\mathcal{X} \subset \mathbb{R}^d$. A sufficient non-collapse condition is $R$ being $L_R$-Lipschitz on $\mathcal{X}$ with $L_R < 1$, resulting in a bi-Lipschitz bound on $F$.

**Definition 1** (Diffusion distance). *For attention matrix $P^{(\ell)}$ at layer $\ell$ and diffusion time $t \geq 1$, define the $t$-step diffusion distance between tokens $i$ and $j$ as*

$$d_{\ell,t}(i, j) \;=\; W_1\big(P^{(\ell)t}(i, \cdot), \, P^{(\ell)t}(j, \cdot)\big),$$

*where $P^{(\ell)t}$ denotes the $t$-th power of the (row-stochastic) attention matrix.*

## S3   METRIC SENSITIVITY.

We rerun the evaluation under alternative ground metrics $d_0$ to test whether measured contraction/topology is specific to positional locality or persists under semantic/syntactic geometries.

**Positional metric.**
$$d_{\text{pos}}(i, j) = \frac{|i - j|}{N - 1}.$$

**Embedding-grounded metric.** Let $e_i \in \mathbb{R}^d$ denote a reference embedding for token $i$ (we use layer-0 hidden states unless stated). Define

$$d_{\text{emb}}(i, j) = \frac{\|e_i - e_j\|_2}{\max_{u,v} \|e_u - e_v\|_2 + \epsilon}.$$

**Cosine-grounded metric.**

$$d_{\cos}(i, j) = \frac{1 - \langle \hat{e}_i, \hat{e}_j \rangle}{\max_{u,v}\big(1 - \langle \hat{e}_u, \hat{e}_v \rangle\big) + \epsilon}, \qquad \hat{e}_i = \frac{e_i}{\|e_i\|_2 + \epsilon}.$$

**Syntactic distance (tree-based).** Let $\text{depth}(\cdot)$ and $\text{LCA}(\cdot, \cdot)$ be computed on a dependency parse of the prompt. Define the (normalized) tree distance

$$d_{\text{syn}}(i, j) = \frac{\text{depth}(i) + \text{depth}(j) - 2\,\text{depth}(\text{LCA}(i, j))}{\max_{u,v}\Big(\text{depth}(u) + \text{depth}(v) - 2\,\text{depth}(\text{LCA}(u, v))\Big) + \epsilon}.$$

If a parse is unavailable for a prompt, we omit this metric for that prompt.

### S3.1   WHAT IS RECOMPUTED UNDER EACH METRIC?

**Curvature.** For each metric $d_0 \in \{d_{\text{pos}}, d_{\text{emb}}, d_{\cos}, d_{\text{syn}}\}$ we compute

$$\widehat{\kappa}(i, j) = 1 - \frac{W_1\big(P(i, \cdot), P(j, \cdot)\big)}{d_0(i, j)}, \qquad \widehat{\kappa}_{\text{lb}} = \text{Quantile}_q\{\widehat{\kappa}(i, j)\},$$

using the same pair-sampling scheme and $q$ as in the main text.

**Direct contraction.** We recompute contraction ratios

$$\widehat{\alpha}_{0.9} = \text{Quantile}_{0.9}\left(\frac{W_1(\mu P, \nu P)}{W_1(\mu, \nu) + \epsilon}\right)$$

under the same random-measure protocol.

**Topology.** We recompute diffusion distances

$$d_{\ell,t}(i,j) = W_1\big(P^{(\ell)t}(i,\cdot), P^{(\ell)t}(j,\cdot)\big)$$

with $W_1$ induced by the chosen $d_0$, and then compute PH summaries on $(\mathcal{T}, d_{\ell,t})$ as in the main text.

**Aggregation.** We report per-metric, per-layer means over prompts (with bootstrap CIs), and we additionally report the layerwise correlation between $\widehat{\kappa}_{\text{lb}}$ and the median/quantile contraction ratio to test the curvature–contraction link under each $d_0$.

**Implementation note.** For $d_{\text{pos}}$ (1D ordered grid) we use the exact $O(N)$ $W_1$ formula. For non-1D metrics we use a consistent approximate solver (Sinkhorn) with fixed regularization; all metrics share the same solver settings to keep comparisons controlled.

## S4   OPTIMAL TRANSPORT AND EXACT $W_1$ ON THE TOKEN LINE

**Dual form and Lipschitz test functions.** For finite metric spaces $(\mathcal{T}, d)$, the Kantorovich–Rubinstein duality gives

$$W_1^d(\mu, \nu) = \sup_{\|f\|_{\text{Lip}(d)} \leq 1} \sum_{i \in \mathcal{T}} f(i)\,(\mu_i - \nu_i), \tag{3}$$

where $\|f\|_{\text{Lip}(d)} = \sup_{i \neq j} |f(i) - f(j)|/d(i,j)$.

**Exact $W_1$ for the 1D positional metric.** When $d = d_{\text{pos}}(i,j) = |i - j|/(N-1)$, $W_1$ admits an exact $O(N)$ formula.

**Lemma S1** (Exact $W_1$ on the discrete line). *Let $\mathcal{T} = \{1, \ldots, N\}$ with $d_{\text{pos}}(i,j) = |i - j|/(N-1)$. Define cumulative sums $F_\mu(k) = \sum_{i \leq k} \mu_i$ and $F_\nu(k) = \sum_{i \leq k} \nu_i$. Then*

$$W_1^{d_{\text{pos}}}(\mu, \nu) = \frac{1}{N-1} \sum_{k=1}^{N-1} \big|F_\mu(k) - F_\nu(k)\big|. \tag{4}$$

*Proof.* This is the standard 1D earth-mover identity on a grid, obtained by writing the optimal transport as moving signed mass along adjacent edges and noting that the minimal cost equals the $\ell_1$ norm of the cumulative imbalance. The normalization $1/(N-1)$ follows from $d_{\text{pos}}(k, k+1) = 1/(N-1)$. $\qquad\square$

**Practical note.** All curvature and contraction experiments under $d_{\text{pos}}$ use equation 4 exactly (no entropic regularization). For non-1D metrics we use Sinkhorn as described in Appendix S6.2.

## S5   FULL PROOFS

### S5.1   PROOF OF THEOREM 1

*Proof of Theorem 1.* Fix $\mu, \nu \in \Delta(\mathcal{T})$ and let $\pi^\star \in \Pi(\mu, \nu)$ be an optimal coupling:

$$W_1(\mu, \nu) = \sum_{x,y} d(x,y)\pi^\star(x,y).$$

For each pair $(x,y)$ choose an optimal coupling $\gamma_{x,y}^\star \in \Pi(P(x,\cdot), P(y,\cdot))$. Define the glued coupling $\Gamma$ on $\mathcal{T} \times \mathcal{T}$ by

$$\Gamma(x', y') = \sum_{x,y} \pi^\star(x,y)\gamma_{x,y}^\star(x', y').$$

A direct marginal check (same as in your draft) shows $\Gamma \in \Pi(\mu P, \nu P)$, hence

$$W_1(\mu P, \nu P) \leq \sum_{x',y'} d(x', y')\Gamma(x', y') = \sum_{x,y} \pi^\star(x, y)\, W_1(P(x, \cdot), P(y, \cdot)).$$

By definition of $\kappa_*$, for all $x \neq y$, $W_1(P(x, \cdot), P(y, \cdot)) \leq (1 - \kappa_*)\, d(x, y)$ (and the $x = y$ case is trivial), so

$$W_1(\mu P, \nu P) \leq (1 - \kappa_*) \sum_{x,y} d(x, y)\pi^\star(x, y) = (1 - \kappa_*)W_1(\mu, \nu),$$

. Iterating results in equation 1. $\hfill \square$

### S5.2   PROOF OF COROLLARY 1

*Proof.* Apply Theorem 1 sequentially:

$$W_1(\mu P_1 \cdots P_L, \nu P_1 \cdots P_L) \leq (1 - \kappa_{*,L})W_1(\mu P_1 \cdots P_{L-1}, \nu P_1 \cdots P_{L-1}),$$

and iterate. $\hfill \square$

### S5.3   PROOF OF THEOREM 2

*Proof.* For all $u, v \in \mathcal{X}$,

$$\|F(u) - F(v)\| = \|(u - v) + (R(u) - R(v))\| \leq \|u - v\| + \|R(u) - R(v)\| \leq (1 + L_R)\|u - v\|.$$

Also,

$$\|F(u) - F(v)\| \geq \big|\|u - v\| - \|R(u) - R(v)\|\big| \geq (1 - L_R)\|u - v\|.$$

If $L_R < 1$ then $\|F(u) - F(v)\| > 0$ for $u \neq v$, so $F$ is injective on $\mathcal{X}$. $\hfill \square$

### S5.4   PROOF OF THEOREM 3

*Proof.* Let $s \sim \mathrm{Unif}(\mathbb{S}^{d-1})$ and write $J = U\Sigma V^\top$. Then $V^\top s \overset{d}{=} s$ and $\mathbb{E}[s_i^2] = 1/d$, so

$$\mathbb{E}\|Js\|^2 = \mathbb{E}\|\Sigma V^\top s\|^2 = \sum_{i=1}^d \sigma_i^2\, \mathbb{E}[s_i^2] = \frac{1}{d}\sum_{i=1}^d \sigma_i^2 = \frac{\|J\|_F^2}{d}.$$

Markov's inequality on $\|Js\|^2$ yields $\mathbb{P}(\|Js\| \geq t) \leq \mathbb{E}\|Js\|^2/t^2 = \|J\|_F^2/(dt^2)$. $\hfill \square$

### S5.5   PROOF OF THEOREM 4

*Proof of Theorem 4.* For any $i, j$,

$$d_{\ell,t}(i, j) = W_1(\delta_i P^{(\ell)t}, \delta_j P^{(\ell)t}) = W_1\big((\delta_i P^{(\ell)})P^{(\ell)(t-1)}, (\delta_j P^{(\ell)})P^{(\ell)(t-1)}\big).$$

Applying Theorem 1 to the kernel $P^{(\ell)}$ for $t - 1$ steps yields

$$d_{\ell,t}(i, j) \leq (1 - \kappa_{\ell,*})^{t-1}\, W_1(\delta_i P^{(\ell)}, \delta_j P^{(\ell)}) = (1 - \kappa_{\ell,*})^{t-1}d_{\ell,1}(i, j),$$

hence $\mathrm{diam}(\mathcal{T}, d_{\ell,t}) \leq (1 - \kappa_{\ell,*})^{t-1} \mathrm{diam}(\mathcal{T}, d_{\ell,1})$. The bound relative to $\mathrm{diam}(\mathcal{T}, d)$ follows similarly from $W_1(\delta_i P^{(\ell)t}, \delta_j P^{(\ell)t}) \leq (1 - \kappa_{\ell,*})^t W_1(\delta_i, \delta_j) = (1 - \kappa_{\ell,*})^t d(i, j)$.

For the lifetime bound: at scale $\varepsilon \geq \mathrm{diam}(\mathcal{T}, d_{\ell,t})$, the Vietoris–Rips complex contains all simplices (it is a full simplex), hence is contractible and has $H_k = 0$ for all $k \geq 1$. Therefore any $k \geq 1$ feature must die by $\varepsilon = \mathrm{diam}(\mathcal{T}, d_{\ell,t})$. $\hfill \square$

### S5.6   WASSERSTEIN LIFTING AS A METRIC FUNCTOR AND THE ROLE OF KERNELS

We now state the functorial property of $W_1$ for deterministic maps and then state the corresponding Lipschitz characterization for kernels.

**Proposition S1** (Non-expansiveness under 1-Lipschitz maps). *Let $f : (X, d_X) \to (Y, d_Y)$ be 1-Lipschitz. Then for all $\mu, \nu \in \Delta(X)$,*

$$W_1^{d_Y}(f_\# \mu, f_\# \nu) \leq W_1^{d_X}(\mu, \nu), \tag{5}$$

*where $f_\#$ denotes pushforward of measures. In categorical terms, $(X, d) \mapsto (\Delta(X), W_1^d)$ defines a functor on $\mathbf{Met}_1$.*

**Theorem S5** (Exact $W_1$-Lipschitz constant of a kernel). *Let $P$ be a Markov kernel on $(X, d)$. Define $\text{Att}_P : \Delta(X) \to \Delta(X)$ by $\mu \mapsto \mu P$. Then*

$$\text{Lip}_{W_1}(\text{Att}_P) = \sup_{x \neq y} \frac{W_1\big(P(x, \cdot), P(y, \cdot)\big)}{d(x, y)} \tag{6}$$

$$= \sup_{x \neq y} \big(1 - \kappa(x, y)\big) = 1 - \inf_{x \neq y} \kappa(x, y). \tag{7}$$

*In particular, if $\inf_{x \neq y} \kappa(x, y) \geq \kappa_* > 0$, then $\text{Att}_P$ is a strict contraction: $W_1(\mu P, \nu P) \leq (1 - \kappa_*) W_1(\mu, \nu)$.*

*Proof of Theorem S5.* Let $C := \sup_{x \neq y} W_1(P(x, \cdot), P(y, \cdot))/d(x, y)$. The gluing argument used above gives, for any coupling $\pi$ of $\mu, \nu$,

$$W_1(\mu P, \nu P) \leq \sum_{x,y} \pi(x, y) W_1(P(x, \cdot), P(y, \cdot)) \leq C \sum_{x,y} \pi(x, y) d(x, y).$$

Taking the infimum over $\pi$ yields $W_1(\mu P, \nu P) \leq C\, W_1(\mu, \nu)$, so $\text{Lip}_{W_1}(\text{Att}_P) \leq C$.

For the matching lower bound, take $\mu = \delta_x$, $\nu = \delta_y$:

$$\frac{W_1(\mu P, \nu P)}{W_1(\mu, \nu)} = \frac{W_1(P(x, \cdot), P(y, \cdot))}{d(x, y)}.$$

Taking the supremum over $x \neq y$ gives $\text{Lip}_{W_1}(\text{Att}_P) \geq C$. Finally, rewrite $C = \sup_{x \neq y}(1 - \kappa(x, y)) = 1 - \inf_{x \neq y} \kappa(x, y)$. $\square$

### S5.7 MULTI-HEAD AND RESIDUAL COMPOSITION

For multi-head attention with concatenation and an output projection $W_O$, apply the above headwise bound and combine via triangle inequality and operator norms. When the attention block is followed by a residual connection, the resulting bi-Lipschitz bounds follow the same structure as in Thm 3, with the contraction term contributed by attention bounded by $(1 - \kappa_{V,*})$ in the $d_V$-geometry.

### S5.8 CIRCULATIONS AND CAUSAL MASKING

**Theorem S6** (Causal masking eliminates directed feedback). *Let $G_{\text{causal}}$ be the directed attention graph induced by a causal (decoder) mask, hence a DAG. Then $G_{\text{causal}}$ admits no nonzero circulation.*

*Proof.* A DAG has no directed cycles, so the claim follows from Lemma S2. $\square$

Theorem S6 is about *directed* circulation on the full masked directed graph. It does *not* assert that an undirected sparsification (e.g. kNN graph) has trivial simplicial $H_1$. Undirected 1-cycles may persist after symmetrization/thresholding even when the directed graph is acyclic and can be detected by persistent homology.

**Causal masking removes directed circulation.** The contraction theorem above concerns metric simplification on measures; it does not by itself explain why decoder-only transformers exhibit a strong architectural bias against *directed feedback loops*. A nonzero *directed circulation* exists if and only if the directed graph contains a directed cycle. Thus, causal masking which makes the attention graph a DAG eliminates directed feedback loops.

**Circulations.** Let $G = (V, E)$ be a directed graph. A circulation is a function $\omega : E \to \mathbb{R}$ such that flow is conserved at every node:

$$\sum_{(u \to v) \in E} \omega(u \to v) \;=\; \sum_{(v \to w) \in E} \omega(v \to w) \quad \text{for all } v \in V.$$

**Lemma S2** (Nonzero circulation $\Leftrightarrow$ directed cycle). *A directed graph admits a nonzero circulation if and only if it contains a directed cycle.*

*Proof.* ($\Rightarrow$) If there is a directed cycle, setting $\omega \equiv 1$ on its edges and $0$ elsewhere is a nonzero circulation.

($\Leftarrow$) Suppose $\omega$ is a nonzero circulation. Pick an edge $e_0 = (v_0 \to v_1)$ with $\omega(e_0) \neq 0$. At $v_1$, conservation implies the total outgoing flow equals total incoming flow; since $e_0$ contributes nonzero incoming flow, there exists an outgoing edge $e_1 = (v_1 \to v_2)$ with $\omega(e_1) \neq 0$. Iterating, we construct a walk $v_0 \to v_1 \to v_2 \to \cdots$ using edges with nonzero flow. Because the graph is finite, some vertex repeats, producing a directed cycle. $\qquad\square$

**Theorem S7** (Causal masking eliminates directed feedback). *Let $G_{\text{causal}}$ be the directed attention graph induced by a decoder causal mask (hence a DAG). Then $G_{\text{causal}}$ admits no nonzero circulation.*

*Proof.* A DAG has no directed cycles. Apply Lemma S2. $\qquad\square$

Theorem S6 is the graph-theoretic statement needed for our empirical directed-cycle diagnostic. It is intentionally separate from any undirected/symmetrized complex used for PH, undirected $H_1$ features can exist even when the directed graph is acyclic.

## S6   ESTIMATORS, ALGORITHMS, AND EXPERIMENTAL SETUP

**Direct Contraction Test.** We apply the operator $\mu \mapsto \mu P$ to random measures drawn from a Dirichlet distribution ($\beta = 0.3$, inducing multimodality) and compute the empirical contraction ratio,

$$\widehat{\alpha}_{0.9}^{(\ell,h)} \;=\; \text{Quantile}_{0.9}\left( \frac{W_1(\mu P, \nu P)}{W_1(\mu, \nu) + \epsilon} \right). \tag{8}$$

$\widehat{\alpha}_{0.9}$ remains bounded by the curvature geometry, providing a direct check of the theoretical bound.

### S6.1   ESTIMATORS REPORTED IN PLOTS/TABLES

Fix a prompt and a layer/head kernel $P^{(\ell,h)}$. We sample $M$ unordered pairs $S = \{(i_m, j_m)\}_{m=1}^{M}$ uniformly from $\{(i, j) : 1 \le i < j \le N\}$.

**Curvature per pair.**

$$\widehat{\kappa}^{(\ell,h)}(i, j) = 1 - \frac{W_1(P^{(\ell,h)}(i, \cdot), P^{(\ell,h)}(j, \cdot))}{d(i, j)}.$$

**Robust curvature lower quantile.**

$$\widehat{\kappa}_{\text{lb}}^{(\ell,h)} = \text{Quantile}_q\left( \{\widehat{\kappa}^{(\ell,h)}(i, j)\}_{(i,j) \in S} \right), \qquad q = 0.05.$$

We also log the sampled minimum $\min_{(i,j) \in S} \widehat{\kappa}(i, j)$ as a conservative check.

**Direct contraction ratios.** We draw $R$ i.i.d. pairs $(\mu_r, \nu_r)$ from Dirichlet($\beta \mathbf{1}$) with $\beta = 0.3$, and compute

$$\alpha_r^{(\ell,h)} = \frac{W_1(\mu_r P^{(\ell,h)}, \nu_r P^{(\ell,h)})}{W_1(\mu_r, \nu_r) + \varepsilon}, \qquad \varepsilon = 10^{-12}.$$

We report $\text{median}(\alpha_r)$ and $\text{Quantile}_{0.9}(\alpha_r)$ per layer/head.

**On-manifold residual stability.** For each $\ell$, from forward passes we collect token states $\{h_{p,i}^{(\ell)}\}$ (prompt $p$, position $i$), sample $K$ pairs $(u,v)$ from this empirical distribution, and compute $\Lambda_\ell(u,v) = \|R_\ell(u) - R_\ell(v)\|/\|u - v\|$. We report $\mathbb{P}(\Lambda_\ell < 1)$ and robust quantiles (5%, 50%, 95%).

### S6.2 EXACT VS. APPROXIMATE $W_1$ IMPLEMENTATIONS

**Positional metric (exact).** For $d = d_{\mathrm{pos}}$, all $W_1$ calls use the exact formula in Lemma S1.

**Non-1D metrics (Sinkhorn).** For embedding/cosine/syntax-derived metrics, we form the cost matrix $C_{ij} = d(i,j)$ and use entropic OT:

$$W_1^\varepsilon(\mu,\nu) = \min_{\pi \in \Pi(\mu,\nu)} \langle \pi, C \rangle - \varepsilon H(\pi),$$

with $\varepsilon \in \{10^{-2}, 10^{-3}\}$ depending on $N$, 200 Sinkhorn iterations, and log-domain stabilization. We validate that conclusions are stable across $\varepsilon$ in Appendix S3.

## S7 RELATED WORK

**Transformers as Dynamic Systems and Operators.** While the original Transformer (Vaswani et al., 2017) was motivated by sequence modeling, recent work interpret the self-attention mechanism through the lens of interacting particle systems and partial differential equations (Chamberlain et al., 2021; Wu et al., 2025). Many other papers interpret the attention matrix as a form of mixing or message passing, where increased depth drives representations toward smoothing or rank collapse phenomena analogous to oversmoothing observed in graph neural networks (GNNs) (Li et al., 2018; Oono and Suzuki, 2020; Dong et al., 2021). We take a complementary geometric perspective by explicitly modeling the attention matrix as a Markov kernel acting on the state space of token indices. This formalism moves beyond spectral analysis of rank collapse (Dong et al., 2021; Anagnostidis et al., 2022) and uses intrinsic geometric invariants to quantify the rate of information concentration.

**Optimal transport and Curvature.** Ollivier's coarse Ricci curvature (Ollivier, 2009) links the geometry induced by a Markov kernel to contraction in Wasserstein distance. Related ideas appear throughout optimal transport (Villani, 2009; Peyré and Cuturi, 2019; Cuturi, 2013) and Markov chain theory, where positive curvature implies rapid mixing and concentration (Joulin and Ollivier, 2010; Paulin, 2016; Eldan et al., 2017). In machine learning, this curvature has been used to analyze community structure in complex networks (Ni et al., 2015; Sandhu et al., 2015; Sia et al., 2019), diagnose oversquashing bottlenecks in GNNs (Topping et al., 2022). While OT has been applied to attention for sparse approximation (Sander et al., 2022), our contribution is to use curvature as a post-hoc diagnostic probe to measure how pretrained LLMs curve the representation space to induce semantic contraction.

**Geometry-aware Transformer architectures.** Another line of work builds geometric structure directly into attention mechanisms using geometric deep learning (GDL) (Bronstein et al., 2021). Recent architectures incorporate explicit equivariance through geometric algebras and physics-informed constraints (de Haan et al., 2024; Spinner et al., 2024) as well as investigate the emergent geometry of deep networks, positing that classifiers work by disentangling data manifolds (Fort and Ganguli, 2019). We build on these views by characterizing transformers as a geometric flow that manipulates the intrinsic dimension (Birdal et al., 2021) and curvature of the data manifold across layers.

**Topological data analysis (TDA) of representations.** Persistent homology provides coordinate-free approach to summarize the multiscale structure of point clouds (Cohen-Steiner et al., 2007; Chazal et al., 2016) and has been used to probe the complexity of deep representations (Rieck et al., 2019; Naitzat et al., 2020b;a). TDA has been successfully employed to analyze decision boundaries (Karimi and Tang, 2020), quantify structural complexity in neural activations (Rieck et al., 2019; Naitzat et al., 2020b), and detect adversarial instability (Zühlke and Kudenko, 2025). Our paper computes topological summaries not on Euclidean distances, but on the diffusion metric induced by the attention operator, directly linking cycle death to the spectral properties of the learning dynamics.

**Hodge theory, diffusion on complexes, and higher-order operators.** Hodge theory generalizes the Laplacian to higher-order structures (edges, triangles), creating a framework for processing flows on simplicial complexes (Schaub et al., 2020; Bunch et al., 2020; Bodnar et al., 2021). Recent work generalizes Hodge decompositions to directed graphs (Jiang et al., 2011), allowing for the separation of gradient flows (potential-driven) from circulations (curl-driven). We apply this decomposition to the causal attention graph, proving that the causal masking constraint structurally forbids harmonic circulations thus enforcing a strict feedforward inductive bias in the message-passing topology.

**Mechanistic interpretability.** Much of the focus here is on reverse-engineering specific computational circuits, such as induction heads or polysemantic neurons (Elhage et al., 2021; Olsson et al., 2022). While these methods identify what functions are computed (e.g. inductive heads, copying), they often struggle to describe global, system-level phenomena. Our work characterizes how the attention mechanism transforms the geometry of the representation space as a global operator.

**High-Dimensional Stability and Concentration.** The stability of deep networks is typically analyzed via the spectral norm of the Jacobian (Bartlett et al., 2017; Miyato et al., 2018). However, recent work suggest that spectral bounds are overly pessimistic and fail to capture typical on-data behavior (Novak et al., 2018), and spectral norm of self-attention can be large but directional gains concentrate well below unity(Liu et al., 2020). We reframe the Lipschitz instability in transformers (Liu et al., 2020; Kim et al., 2021) as a concentration phenomenon rather than a fundamental failure of the architecture.

## S8 EMPIRICAL VERIFICATION: OPERATOR CONTRACTION AND TOPOLOGICAL SIGNATURE

Sections S2–2 make three testable predictions regarding the geometry of representations. To test these, we implement a model-agnostic probe that estimates curvature, directly measures $W_1$ contraction on random measures, quantifies on-data residual stability, and summarizes undirected topology via PH.

**Models and data.** We evaluate `gpt2` and `gpt2-medium` (causal/decoder-only) to test the contraction and masking hypotheses. We employ `bert-base-uncased` (bidirectional) as a counterfactual baseline for the directed-cycle diagnostic. Results are reported over $P = 50$ natural-text prompts with a fixed sequence length of $N = 128$, utilizing standard padding and truncation.

### S8.1 CURVATURE–CONTRACTION–STABILITY–TOPOLOGY PROBE

**Token set and base metric.** For a prompt with tokens $T = \{1, \ldots, N\}$, we construct the base metric space $(T, d_0)$. In our primary experiments, $d_0$ is the normalized positional distance $d_0(i, j) = \frac{|i-j|}{N-1}$, $i, j \in T$. We treat the attention matrix $A^{(\ell,h)}$ (post-softmax) as a Markov transition kernel $P^{(\ell,h)}$ on this metric space. To ensure robustness, we replicate the full analysis using semantic and syntactic ground metrics in Appendix S3.

**Exact $W_1$ on the 1D token line.** Since $(T, d_0)$ is an ordered 1D grid, the 1-Wasserstein distance between $p, q \in \Delta(T)$ admits an exact $O(N)$ formula via cumulative sums $W_1(p, q) = \frac{1}{N-1} \sum_{k=1}^{N-1} |F_p(k) - F_q(k)|$. This removes entropic/Sinkhorn approximation error and makes curvature and contraction estimation numerically stable.

**Robust curvature lower bound.** For each $(\ell, h)$ and each sampled pair $(i, j)$ with $i \neq j$, we compute coarse curvature

$$\widehat{\kappa}^{(\ell,h)}(i, j) = 1 - \frac{W_1\big(P^{(\ell,h)}(i, \cdot), \, P^{(\ell,h)}(j, \cdot)\big)}{d_0(i, j)}. \tag{9}$$

Curvatures can be heavy-tailed across pairs, so we summarize by a *robust lower quantile* $\widehat{\kappa}_{\mathrm{lb}}^{(\ell,h)} :=$ $\mathrm{Quantile}_q\Big(\{\widehat{\kappa}^{(\ell,h)}(i, j)\}_{(i,j)\sim S}\Big)$, $q = 0.05$, where $S$ is a uniform sample of $M$ pairs (default $M = 5{,}000$ per layer/head). We also report mean/median curvature, but $\widehat{\kappa}^{(\ell,h)}$is the statistic aligned with Thm. 1.

**On-data residual stability.** Here, we quantify stability on the data manifold rather than in the worst case. For each layer $\ell$, we sample hidden states $u, v \sim p_\ell$ from forward passes on the prompt set, and compute the on-data directional Lipschitz statistic,

$$\Lambda_\ell(u, v) = \frac{\|R_\ell(u) - R_\ell(v)\|}{\|u - v\|}.$$

We report probability of stability $\mathbb{P}(\Lambda_\ell < 1)$ and robust quantiles of $\Lambda_\ell$, and compare them to Cor. 2.

**Directed-cycle diagnostic (masking).** We verify Thm. S6 by constructing a thresholded graph $G_\tau$ where $(i \rightarrow j)$ exists if $P_{ij} \geq \tau$. We apply a topological sort to detect directed cycles. We predict strictly zero cycles for GPT-2 (causal) and non-zero cycles for BERT (bidirectional).

**Undirected PH.** To measure the complexity of the diffusion geometry, we compute the PH of the space $(T, d_t^{(\ell,h)})$, where $d_t$ is the diffusion distance at time $t$. We summarize the topology using the lifetime of the most persistent features using Vietoris–Rips complex in dimension 1 (loops) $\widehat{\tau}_1^{(\ell,h)} := \max_k(\text{death}_k - \text{birth}_k)$. We use two signatures $H_0$ persistence (cluster longevity) and maximum $H_1$ lifetime (largest cycle). Note, while directed cycles are forbidden, undirected loops will persist in early layers but collapse ($\widehat{\tau}_1 \rightarrow 0$) in deep layers as the diffusion geometry contracts.

## S8.2 Controls and Robustness

**Aggregation and uncertainty.** All curves are aggregated over prompts (and optionally over heads). We report promptwise means with $95\%$ bootstrap confidence intervals (resampling prompts) for $\widehat{\kappa}_{\text{lb}}$, $\widehat{\alpha}_{\text{med}}$, $\widehat{\alpha}_{0.9}$, $\mathbb{P}(\Lambda_\ell < 1)$, and $\widehat{\tau}_{k,0.9}$.

**Sensitivity analyses.** We vary diffusion time $t$, Dirichlet concentration $\beta$, sequence length $N$, token subsampling strategy for PH, and the scale grid $\mathcal{E}$. We also rerun the entire pipeline under alternative ground metrics $d_0$ (Appendix S3) to test whether the contracted geometry aligns with positional locality versus semantic similarity.

**Computational complexity.** Using equation 4, each $W_1$ evaluation is $O(N)$. Curvature estimation is $O(MN)$ per layer/head. PH is computed on $N \leq 128$ vertices and remains tractable; full implementation details are in Appendix S6.

## S8.3 Dataset, prompts, and model details

**Models.** We use `gpt2` and `gpt2-medium` (decoder-only) and `bert-base-uncased` (bidirectional) via HuggingFace Transformers. We extract *post-softmax* attention weights and (unless otherwise stated) average attention over heads to form $P^{(\ell)}$.

**Prompts and length.** We evaluate $P = 50$ natural-text prompts (English), using fixed length $N = 128$ via truncation/padding. All statistics are aggregated over prompts; bootstrap CIs resample prompts.

**Causal masking checks.** For GPT-style models, we report: (i) forbidden (future) mass $\sum_i \sum_{j>i} P_{ij}$ (should be $\approx 0$), (ii) directed-cycle presence in a thresholded graph $G_\tau$ where edges are $(i \rightarrow j)$ with $P_{ij} \geq \tau$, checked by topological sorting. For BERT, the same cycle check is applied (cycles typically appear).

## S8.4 Ablations

All ablations preserve the causal support (i.e., $P_{ij} = 0$ for $j > i$) when applied to GPT-style models.

- **Uniform:** for each row $i$, set $P_{ij} \propto \mathbf{1}[j \leq i]$.
- **Random-stochastic:** for each row $i$, sample $(P_{i1}, \ldots, P_{ii}) \sim \text{Dirichlet}(\mathbf{1})$ and set $P_{ij} = 0$ for $j > i$.
- **Positional decay:** set $P_{ij} \propto \exp(-|i - j|/\tau)\mathbf{1}[j \leq i]$ with $\tau$ tuned so row entropy roughly matches the trained kernel.

- **Permuted keys:** within each row $i$, permute the key indices $j \leq i$ (equivalently, permute columns restricted to causal support) and renormalize.

### S8.5 PERSISTENT HOMOLOGY CONFIGURATION

For each layer $\ell$, we compute a full $N \times N$ distance matrix $D^{(\ell)}$ on $N = 128$ points. We compute Vietoris–Rips persistence (dimensions 0 and 1) and summarize by, (i) $H_0$ total persistence / number of long-lived components, and (ii) max $H_1$ lifetime $\max_k(\mathrm{death}_k - \mathrm{birth}_k)$. For numerical stability we cap distances to $[0, 1]$ (under normalized metrics) and deduplicate exact zeros.

### S8.6 COMPUTE AND REPRODUCIBILITY

We use $M = 5{,}000$ curvature pairs per layer, $R = 200$ Dirichlet contraction trials per layer, diffusion time $t = 2$, curvature quantile $q = 0.05$, and random seed fixed per prompt. We provide code to recompute all plots from cached attention tensors and logged summaries.

