# OpenReview forum: "The Geometrical and Topological Signature of Transformers"
_ICLR.cc/2026/Workshop/GRaM — ICLR 2026 Workshop GRaM Poster_

### Official Review · Reviewer_jVbd · 2026-02-21
**Intriguing Geometric Framework, but Needs Modern Validation**

**Rating:** 5
**Confidence:** 2

**Review:**

### Summary
The authors propose a topological framework to analyze the layerwise evolution of transformer representations by modeling attention heads as Markov kernels on a token metric space. This formulation admits a Wasserstein-1 ($W_{1}$) lifting where coarse Ollivier-Ricci curvature provides quantitative bounds on the action of the induced operator. The paper demonstrates that positive curvature implies layerwise Wasserstein contraction, while negative curvature implies expansion. Empirically, the authors introduce a reproducible probe to estimate curvature and track topological simplification using persistent homology. Experiments on GPT-2 models show a transition from expansive to contractive regimes, where topological complexity (measured by $H_{1}$ lifetimes) collapses in deeper layers. The paper also provides theoretical bounds on residual connections, arguing they prevent total feature-space collapse by maintaining bi-Lipschitz stability.

### Strengths
* **Novel Theoretical Framework:** The paper offers a rigorous geometric and topological perspective on transformers, successfully linking attention mechanisms to Markov chains and optimal transport theory.
* **Strong Mathematical Rigor:** The theoretical contributions are significant, particularly Theorem 1 (connecting curvature to $W_{1}$ contraction) and Theorem 2 (providing conditions for residual stability). The use of persistent homology to quantify the "annealing" of topological complexity is a creative and mathematically grounded approach.
* **Clear Correlation:** The authors demonstrate a strong empirical correlation between their theoretical curvature estimates and the observed contraction of representations ($\rho \approx -0.8$ for GPT-2), validating their geometric hypothesis.

### Weaknesses
* **Outdated and Limited Models:** The empirical validation is restricted to GPT-2 (small and medium) and BERT. In the context of 2026, evaluating exclusively on models from 2018-2019 is a significant limitation. The findings would be much more compelling if validated on modern, larger-scale architectures (e.g., Llama, Mistral) to ensure the observed geometric phase transition is not an artifact of older architectures.
* **Restricted Experimental Scope:** The experiments use a very small sample size (50 prompts) and a short fixed sequence length (128 tokens). Given that the paper discusses "diffusion geometry" and long-range dependencies, testing on such short contexts limits the robustness of the conclusions regarding topological collapse.
* **Clarity and Readability:** The paper is dense and difficult to follow for readers not deeply versed in algebraic topology and optimal transport. The transition between theoretical definitions (like Vietoris-Rips filtration) and their practical implications for model performance or interpretation is often abrupt. The paper would benefit significantly from more intuitive explanations of *why* specific topological features (like $H_{1}$ lifetimes) matter for language modeling capability.
* **Presentation and Formatting:** Visual and structural choices further hinder readability. The text within the figures (particularly Figures 1 and 2) is exceptionally small and difficult to read without significant zooming. Additionally, relegating the mathematical notations to the appendix forces a disjointed reading experience, as the reader must constantly jump back and forth to parse the core claims in the main text.

### Overall Recommendation
While the theoretical framework is novel and mathematically sound, the empirical validation is severely lacking for a current conference submission. The reliance on GPT-2 and very short context windows makes it difficult to assess the generality of the claims for modern Large Language Models. Currently, the gap between the sophisticated theory and the limited experiments is too large.

**Pmlr Suitability:**

NA

---

### Meta-Review · Area_Chair_wHM5 · 2026-02-26

**Decision:**

Accept

**Metareview:**

This paper studies how token representations evolves through successive layers. The main idea is to treat each attention head as a Markov kernel, and study their concentration in distribution using the Ollivier-Ricci curvature. It gives some topological and geometrical insights. We accept this paper.

**Relevance To Proceedings:**

Tiny paper — does not apply

**Relevance To Workshop:**

Yes — suitable for GRaM

---

### Decision · Program_Chairs · 2026-03-02

Accept (Poster)